# Expansion of a neural crest gene signature following ectopic MYCN expression in sympathoadrenal lineage cells in vivo

**Rodrigo Ibarra-García-Padilla[1,2], Annika Nambiar[1], Thomas A. Hamre[1], Eileen W. Singleton[1], Rosa A. Uribe[1]***

1 Department of Biosciences, Rice University, Houston, Texas, United States of America, 2 Biochemistry and Cell Biology Graduate Program, Rice University, Houston, Texas, United States of America

* rosa.uribe@rice.edu

## Abstract

Neural crest cells (NCC) are multipotent migratory stem cells that originate from the neural tube during early vertebrate embryogenesis. NCCs give rise to a variety of cell types within the developing organism, including neurons and glia of the sympathetic nervous system. It has been suggested that failure in correct NCC differentiation leads to several diseases, including neuroblastoma (NB). During normal NCC development, MYCN is transiently expressed to promote NCC migration, and its downregulation precedes neuronal differentiation. Overexpression of MYCN has been linked to high-risk and aggressive NB progression. For this reason, understanding the effect overexpression of this oncogene has on the development of NCC-derived sympathoadrenal progenitors (SAP), which later give rise to sympathetic nerves, will help elucidate the developmental mechanisms that may prime the onset of NB. Here, we found that overexpressing human EGFP-MYCN within SAP lineage cells in zebrafish led to the transient formation of an abnormal SAP population, which displayed expanded and elevated expression of NCC markers while paradoxically also co-expressing SAP and neuronal differentiation markers. The aberrant NCC signature was corroborated with *in vivo* time-lapse confocal imaging in zebrafish larvae, which revealed transient expansion of *sox10* reporter expression in MYCN overexpressing SAPs during the early stages of SAP development. In these aberrant MYCN overexpressing SAP cells, we also found evidence of dampened BMP signaling activity, indicating that BMP signaling disruption occurs following elevated MYCN expression. Furthermore, we discovered that pharmacological inhibition of BMP signaling was sufficient to create an aberrant NCC gene signature in SAP cells, phenocopying MYCN overexpression. Together, our results suggest that MYCN overexpression in SAPs disrupts their differentiation by eliciting abnormal NCC gene expression programs, and dampening BMP signaling response, having developmental implications for the priming of NB *in vivo*.

**Data Availability Statement:** All relevant data are within the paper and its Supporting information files.

**Funding:** This work was funded by Cancer Prevention and Research Institute of Texas (CPRIT) grant RR170062 and National Science Foundation (NSF) grant 1942019 awarded to R.A. U. The funders had no role in study design, data collection and analysis, decision to publish, or preparation of the manuscript.

**Competing interests:** The authors have declared no competing interests.

# Introduction

Neural crest cells (NCC) are a transient, multipotent, and highly migratory stem cell population present during early vertebrate embryogenesis. NCCs are born from neuroepithelial precursors along the dorsal neural tube, the transient structure that gives rise to the central nervous system, and are spatially subdivided along the anterior-posterior embryo length into cranial, vagal, trunk, and sacral populations [1–4]. All NCCs undergo an epithelial-to-mesenchymal transition (EMT) to delaminate from the neural tube and migrate extensively throughout the developing body [3, 5–8]. NCCs differentiate into diverse cell types, depending on the anteroposterior axial level they arose from, to become cellular components of many critical tissues—including craniofacial cartilage, muscle, bone, pigment cells of the skin, and peripheral nervous system ganglia, such as enteric and sympathetic ganglia [2, 4, 9–12]. Across their development, NCCs present extremely dynamic transcriptional programs that are tightly regulated to dictate their final fate acquisition. These complex transcriptional landscapes, required for distinct cell fate acquisition, have been described with high detail through gene regulatory networks during early NCC development [13–17].

One major derivative of the NCC is the sympathetic nervous system [18, 19]. The sympathetic nervous system, a subdivision of the autonomic nervous system, innervates internal organs, smooth muscle, and exocrine glands in vertebrates and is essential for organ homeostasis. Sympathetic neurons function to control heart rate, body temperature, and endocrine secretion in response to external stimuli [20, 21]. Strong characterization of gene regulatory networks has been fundamental to understanding the transcriptional control that is required for correct sympathetic differentiation [10, 15, 18, 22]. In particular, vagal and trunk NCC subpopulations, which hail from the neuroaxis adjacent to somite levels 1–28, give rise to sympathetic neurons and glia, and to chromaffin cells in the adrenal medulla [9, 10, 12, 18–20, 23]. During early NCC development, the transcription factors FoxD3, Tfap2a, and Sox9, which are important for NCC lineage specification, regulate the expression of Sox10, which is a transcription factor widely accepted to be expressed by migrating NCCs and NCC-derived fates [10, 15, 24]. In zebrafish, in addition to *sox10*, the *crestin* marker is expressed in premigratory and migratory NCCs, and its expression is gradually downregulated in differentiating cells [25].

To begin positioning into the future sites of sympathetic ganglia, NCCs activate Snai2 and Twist1 to promote EMT and migrate from the neural tube ventrally towards the dorsal aortae, embryonic vessels that later progress to form the descending aorta [8, 10, 19, 26]. Once NCCs reach the dorsal aorta, they become known as sympathoadrenal precursors (SAPs), a transitionary progenitor state [9, 10, 12, 18, 20]. SAPs express the transcription factor Phox2b, among other sympathetic differentiation factors, while downregulating Sox10 [27]. Once specified, SAPs begin to differentiate into neurons expressing adrenergic-lineage enzymes like Tyrosine Hydroxylase (TH) and Dopamine β-hydroxylase (Dbh) that are involved in catecholamine synthesis––such as dopamine, epinephrine, and norepinephrine, which are neurotransmitters mainly produced by the adrenal gland and the sympathetic nervous system [10, 12, 15, 18, 28]. In addition, the expression of the SAP network of transcription factors, which dictates NCC differentiation towards a sympathetic fate, is induced as the cells receive cues from the surrounding tissue microenvironment. One of the major extrinsic factors SAPs receive that direct their migration and final cell fates include the Bone Morphogenetic Proteins (BMPs); particularly BMP2, BMP4, and BMP7 have been shown to induce sympathetic differentiation of NCCs [29–31].

NB is a devastating pediatric cancer characterized by the formation of solid tumors in the adrenal glands and in the para-spinal sympathetic ganglia along the abdomen, chest, neck, and

pelvis [32–34]. High-risk NB patients exhibit less than 50% survival chance even when they are submitted to a variety of treatments [33, 35, 36]. Because more than half of NB patients are classified into the high-risk group, understanding its pathological mechanism is of utmost importance. Towards this end, much attention has centered on the oncogene MYCN, which has been recognized as an important driver of NB tumorigenesis and as a marker for poor prognosis within high-risk patients [34, 37–39]. Multiple animal models have shown that over-expressing MYCN in the SAP population leads to tumorigenesis with histological similarities to NB patients [40–43]; however, many of these studies did not focus on the early development of the pathology. During development, MYCN is expressed in emigrating NCCs, and its down-regulation precedes neuronal differentiation [44]. Tissue-restricted knockdown of MYCN in the neural progenitor and NCC populations showed increased neuronal differentiation [45]. Additionally, mouse embryos with loss of MYCN present an underdeveloped nervous system, particularly evident in reduced cellularity of cranial and spinal ganglia [46]. Despite this knowledge, the effect dysregulated MYCN expression has on NCC development, particularly when undergoing SAP differentiation, has not been fully characterized *in vivo*.

In this work, we leverage the vertebrate model zebrafish, which has been a staple for cancer research for over a decade [47], to characterize the early role MYCN overexpression plays in NCC to sympathetic development. We used the zebrafish MYCN-driven NB model, which expresses EGFP-tagged human MYCN within SAPs [43]. Here, we present evidence that MYCN overexpression *in vivo* elicits aberrant NCC gene expression signatures in SAPs transiently, while also dampening BMP signaling. Together, our results have implications for the developmental priming of NB.

## Results

### MYCN overexpression does not affect early sympathoadrenal (SAP) cell numbers

Given the reported roles of the MYC transcription factor family in several physiological processes within the cell, such as proliferation, differentiation, and survival [39], we first aimed to determine if MYCN overexpression within the SAP lineage affected cell numbers during early sympathetic development. Towards this end, we used the previously characterized zebrafish line *dbh*:EGFP-MYCN [43], which expresses human EGFP-MYCN within SAP lineage cells, or *dbh*:EGFP [43], expressing EGFP alone, as a control. We focused on the superior cervical ganglia (SCG), the anterior-most and first-formed sympathetic cells within the developing sympathetic ganglion chain [48]. While *dbh* transcript is first detected at the onset of 2 days post fertilization (dpf) [48], the earliest we could reliably identify and sort for embryos expressing *dbh*-driven EGFP/EGFP-MYCN within their SCG was at the transition between 2 and 3 dpf. Therefore, we focused our analyses on larvae beginning at 3 dpf. To quantify SAP numbers within the SCG, our region of interest (ROI) (Fig 1A), we used IMARIS to identify individual cells from 3 to 6 dpf (Fig 1C–1L), across 3D datasets (S1 Movie). We quantified the total number of cells expressing either EGFP or EGFP-MYCN (Fig 1B). At 3 dpf control larvae presented an average of 13.73 EGFP+ cells compared to 12.14 EGFP-MYCN+ cells in MYCN overexpressing larvae. As larvae continued developing and the sympathetic ganglion chain expanded, the SCG cell numbers increased by a slight amount, where at 4 dpf the mean cell number in control larvae was 15.53, and in MYCN 16.53. After these timepoints, the SCG cell numbers remained similar between the two conditions at 5 dpf (17.92 vs 20.22) and 6 dpf (17.33 vs 17.29). Overall, we did not detect any significant changes in SAP numbers in the SCG between EGFP+ and EGFP-MYCN+ larvae. These data indicate that MYCN overexpression does not alter SCG SAP population size during these early sympathetic differentiation

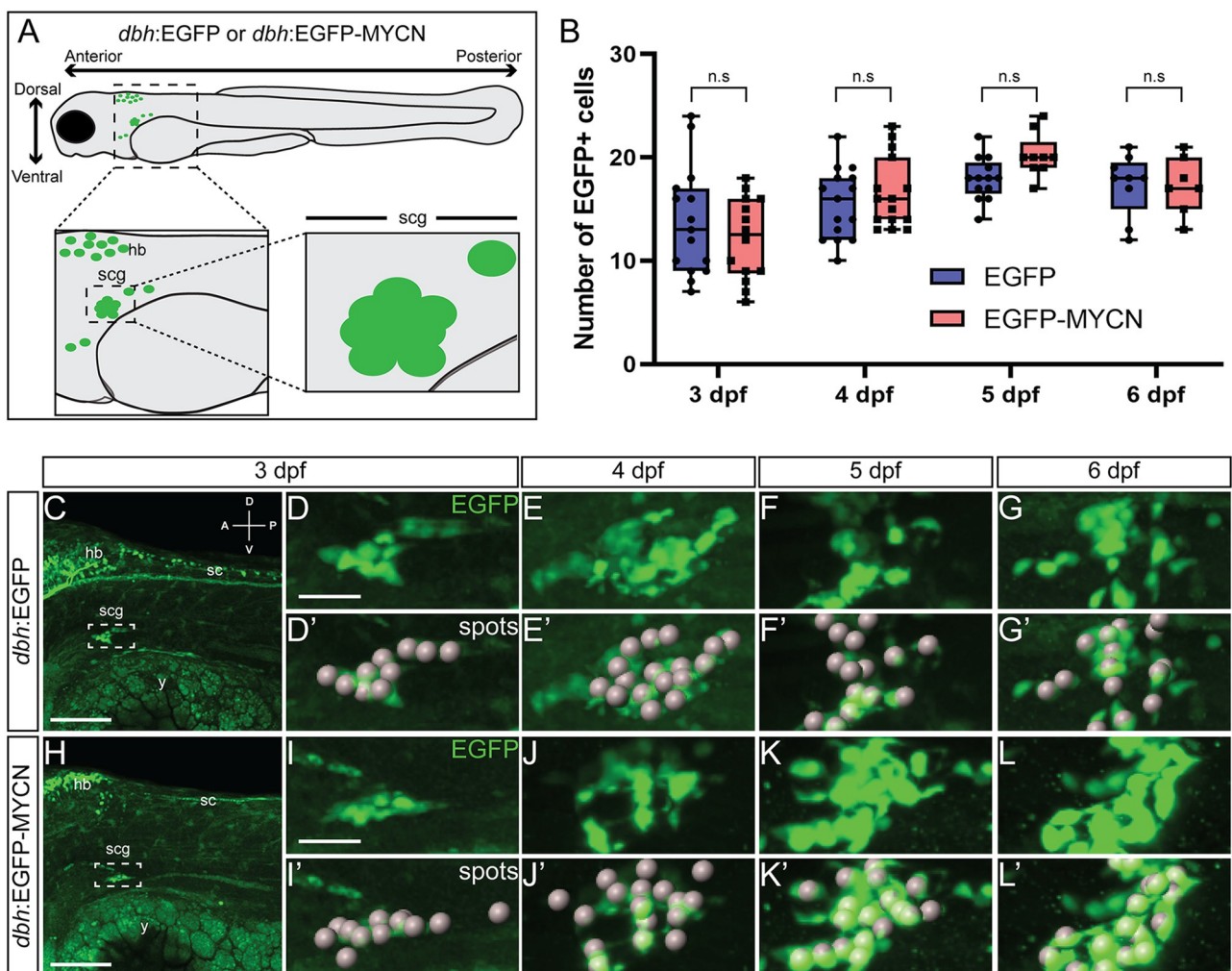

**Fig 1. MYCN overexpression in SAP cells does not increase their cell numbers during early larval stages.** A) Cartoon model of a 3 dpf *dbh*:EGFP or *dbh*:EGFP-MYCN larval fish, depicting imaging regions and selection of the SCG as the region of interest (ROI). B) Quantification of cells detected at 3, 4, 5, and 6 dpf after "spots" detection using IMARIS. For 3 dpf EGFP n = 15, EGFP-MYCN n = 14; for 4 dpf EGFP n = 15, EGFP-MYCN n = 15; for 5 dpf EGFP n = 13, EGFP-MYCN n = 9; and for 6 dpf EGFP n = 9, EGFP-MYCN n = 7. C-L) Representative images of *dbh*:EGFP (C-G) or *dbh*: EGFP-MYCN (H-L) larvae from 3 to 6 dpf. Boxed region represents the SCG. ROI from 3, 4, 5, and 6 dpf larvae is shown in full detail (D-G, H-L), with the cells detected after performing the spots pipeline in IMARIS (D'-G', H'-L'). A (anterior), P (posterior), D (dorsal), V (ventral) axes shown in upper right corner. hb = developing hindbrain, sc = developing spinal cord, cg = developing cranial ganglia, scg = developing superior cervical ganglia, y = yolk. Scale bars = 100 μm for full size image, 25 μm cropped images. n.s., non-significant (P>0.05).

timepoints. However, this does not rule out that MYCN could be playing a role in early development of this population.

## Ectopic MYCN expression in SAP cells leads to an expanded neural crest cell gene expression signature

Using *sox10*-derived single-cell transcriptomic zebrafish datasets, that capture early NCC differentiation at 24, 48–50, and 68–70 hours post fertilization (hpf) [49, 50], we queried for the expression of NCC markers *crestin* and *sox10* along these developmental time points (Fig 2A). For simplicity, we refer to the 48–50 hpf time as simply 48 hpf, and the 68–70 hpf as simply 69

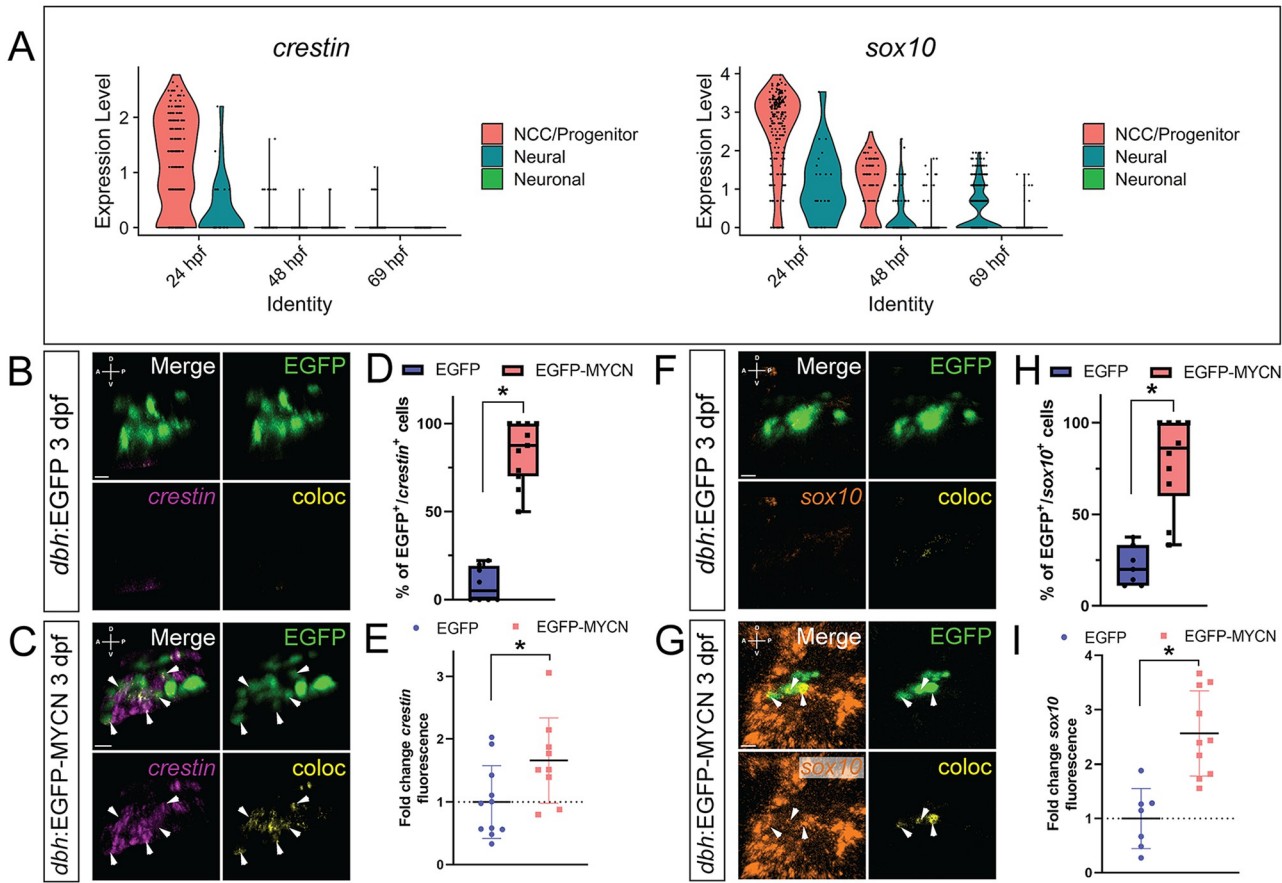

**Fig 2. MYCN overexpressing SAP cells display an ectopic NCC gene expression signature.** A) Violin plots depicting *crestin* (left) and *sox10* (right) expression from single cell datasets of *sox10* derived cells at 24, 48, and 69 hpf. B,C,F,G) WICHCR performed using HCR probes against *crestin* (B,C) or *sox10* (F,G) and with antibody against EGFP on 3 dpf *dbh*:EGFP (B,F) and *dbh*:EGFP-MYCN (C,G) larvae. Representative images of the SCG region reveal expression of the markers and their colocalized channel (coloc). D,H) Percentage of EGFP+ or EGFP-MYCN+ cells co-expressing *crestin* (D) or *sox10* (H) in 3 dpf larvae. For crestin EGFP n = 8, EGFP-MYCN n = 11; for sox10 EGFP n = 7, EGFP-MYCN n = 10. E,I) Mean *crestin* (E) or *sox10* (I) fluorescence intensity quantified in the SCG and normalized to *dbh*:EGFP average intensity at 3 dpf. For *crestin* EGFP n = 11, EGFP-MYCN n = 8; for *sox10* EGFP n = 7, EGFP-MYCN n = 10. Markers: EGFP (green), *crestin* (magenta), *sox10* (orange) and coloc channel (yellow). A (anterior), P (posterior), D (dorsal), V (ventral) axes shown in upper left corner. Scale bars = 10 μm. For all graphs * P<0.05.

hpf in Fig 2. We found that expression of these transcripts was highly enriched during early timepoints of NCC differentiation. In particular, we found that *crestin* was mostly expressed within NCC at 24 hpf but its levels rapidly reduced at later time points (Fig 2A). In addition, *sox10* was enriched in NCC, neural, and neuronal fated cells at 24 and 48 hpf, with relative lower *sox10* transcript expression in neural fated cells at 69 hpf. In avian models, *mycn* is expressed during early neural tube and NCC development, conversely, *mycn* is downregulated in sympathetic cells that form the peripheral nervous system [44, 51]. Furthermore, in zebra-fish, *mycn* expression has been shown in NCC at 16 hpf and 24 hpf, however, *mycn* was largely absent in NCC-derived neuronal fated cells at later time points [52].

We assessed changes in *mycn* levels over time in the developing zebrafish embryo and larval fish. Single cell analysis showed higher relative *mycn* expression in early progenitor populations when compared with later populations (S1A Fig). Whole mount *in situ* hybridization showed the spatiotemporal expression pattern of *mycn* in zebrafish larvae at 48 hpf and 70 hpf (S1B and S1C Fig). *mycn* was expressed in several anatomical regions (brain, developing eyes,

gut region, branchial arches) at 48 hpf (S1B Fig). Conversely, at 70 hpf the transcript became more spatially restricted in the midgut, developing eye and brain (S1C Fig). These results are in accordance with previous reports indicating *mycn* expression during early development, with reduced levels as development progresses [44, 53]. Together, these data indicate that *mycn* is expressed primarily during earlier time points when NCC are multipotent, migratory and in the early stages of differentiation.

Since ectopic MYCN expression in the early SCG did not affect cell numbers (Fig 1), we wondered if MYCN overexpression could affect SAP cell differentiation. To that end, we used Whole-mount Immuno-Coupled Hybridization Chain Reaction (WICHCR) [54] to analyze gene expression of the NCC marker transcripts *crestin* (Fig 2B–2E, S2 Fig) and *sox10* at 3 dpf (Fig 2F–2I), and 4 and 5 dpf (S2 Fig). Within the vicinity of the SCG, we found a qualitative increase in both of these NCC markers in EGFP-MYCN$^+$ larvae, when compared with EGFP$^+$ controls. Particularly, we observed an expansion of *crestin* and *sox10* expression domains in MYCN overexpressing larvae at 3 dpf when compared with control (Fig 2B, 2C, 2F and 2G), that continued at 4 dpf and then decreased by 5 dpf (S2 Fig).

We next counted how many of the EGFP-MYCN$^+$, or EGFP$^+$ cells, were co-positive for each of the NCC markers by detecting co-localization between EGFP$^+$ and *crestin* or *sox10*. We detected a statistically significant increase in the percentage of double positive cells (EGFP$^+$/*crestin*$^+$), when comparing EGFP-MYCN$^+$ larvae with controls at 3 dpf (p<0.000001) (Fig 2D), 4 dpf (p = 0.001521), and 5 dpf (p<0.000001) (S2 Fig). *sox10* expressing cells were also significantly increased in EGFP-MYCN$^+$ larvae, when compared to controls at 3 dpf (p = 0.000048) (Fig 2H), 4 dpf (p = 0.000182), and 5 dpf (p = 0.015749) (S2 Fig). To determine whether *crestin* and *sox10* were differentially expressed in the EGFP-MYCN$^+$ larvae when compared with EGFP$^+$ larvae, mean *crestin* or *sox10* fluorescence intensity was calculated in the developing SCG at 3 dpf (Fig 2E and 2I). This analysis showed that EGFP-MYCN$^+$ larvae displayed significantly higher levels of *crestin* (Fig 2E, p = 0.030554) and *sox10* (Fig 2I, p = 0.000391) compared with control larvae. These data show that MYCN overexpression in the SAP lineage leads to aberrant gene expression where a higher percentage of SAPs express NCC markers past their normal expression windows. Additionally, the levels of expression for these NCC markers are higher within the region of the developing SCG.

## MYCN overexpression transiently expands *sox10* reporter signal *in vivo*

Taking advantage of the powerful imaging techniques available for zebrafish, we sought to characterize the expression dynamics of the NCC marker *sox10 in vivo*. For this, we crossed the *dbh*:EGFP or *dbh*:EGFP-MYCN fish lines with the *sox10*:mRFP fish line that produces membrane bound RFP, as a reporter, in *sox10* expressing cells [55], and performed *in vivo* time-lapse imaging starting at ~70 hpf, during early SAP differentiation, and ending at ~118 hpf. Qualitatively, we observed that mRFP signal in the SCG at the beginning of the timelapse was higher in EGFP-MYCN$^+$ larvae (Fig 3B–3B''', S7–S11 Movies), while the EGFP$^+$ controls presented minimal mRFP signal in that region (Fig 3A–3A''', S2–S6 Movies). When we analyzed mRFP levels at 90 hpf, EGFP$^+$ larvae presented no expression of mRFP in the SCG (Fig 3C–3C''', S2–S6 Videos), whereas EGP-MYCN$^+$ continued to present mRFP signal (Fig 3D–3D''', S7–S11 Videos), albeit lower than at the beginning of the timelapse. Later, the expression of mRFP by EGFP-MYCN$^+$ cells was drastically reduced by 5 dpf (Fig 3F). These data indicate that *sox10* reporter expression in EGFP-MYCN$^+$ larvae is expanded transiently from 3 to 5 dpf, in agreement with our observations of *sox10* transcript (Fig 2, S2 Fig).

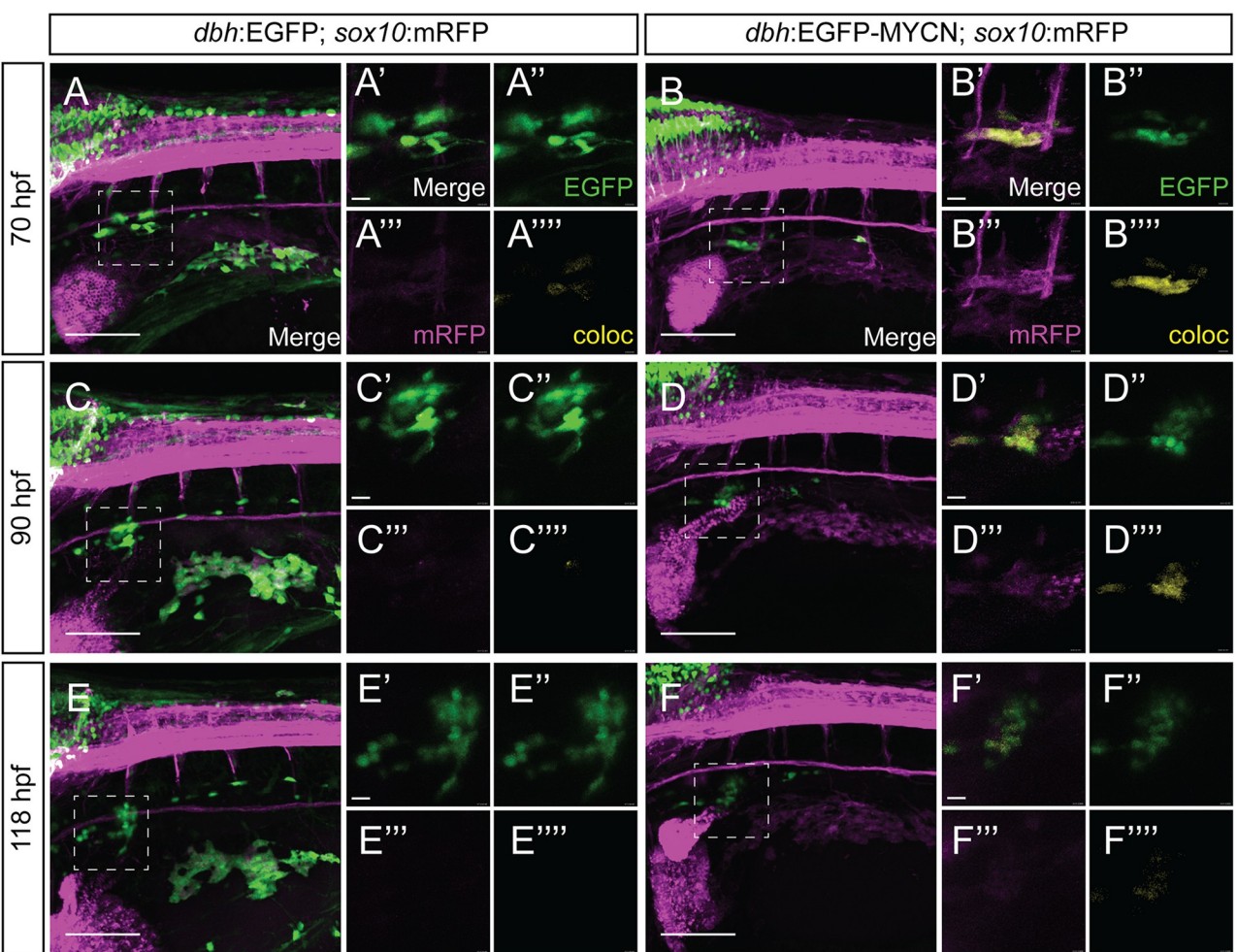

**Fig 3. MYCN overexpression in SAP cells leads to *sox10* reporter activity expansion *in vivo*.** A-F) Confocal images of the developing SCG from different time points during live imaging of *sox10*:mRFP;*dbh*:EGFP (A,C,E), or *sox10*:mRFP;*dbh*:EGFP-MYCN (B,D,F) larvae from ~70 hpf to ~118 hpf. Markers: EGFP (green), *sox10* reporter: mRFP (magenta), and coloc channel (yellow). *sox10* reporter expression of mRFP within the developing SCG is visible in MYCN-EGFP overexpressing larvae (B',D', F') compared to EGFP control larvae, (A', C', E') with higher reporter levels at earlier timepoints. Scale bars = 50 μm for uncropped images, 20 μm for cropped images.

## Neuronal differentiation marker gene expression is not affected by ectopic MYCN expression in SAP cells

The expanded NCC marker expression by EGFP-MYCN⁺ cells suggests that ectopic MYCN may lead to an aberrant NCC-like population that fails to maintain correct SAP identity. To decipher this, we performed WICHCR to detect SAP differentiation markers *phox2bb* and *dbh* [56, 57] at 3 to 5 dpf, time points where these genes are expected to be expressed by sympathetic fated cells [49]. We found these SAP markers to be expressed by both EGFP⁺ and EGFP-MYCN⁺ larvae from 3–5 dpf (Fig 4A–4H, S3 Fig). After quantifying the percentage of double positive (EGFP⁺/*phox2bb*⁺) cells, we observed no significant changes at 3 dpf (Fig 4C, p = 0.884052), 4 dpf (p = 0.079465), and 5 dpf (p = 0.557821) (S3 Fig), when comparing EGFP⁺ with EGF-MYCN⁺ larvae. *dbh* expressing cells also remained similar between EGFP⁺ and EGFP-MYCN⁺ larvae at 3 dpf (Fig 4G, p = 0.89812). After analyzing *phox2bb* and *dbh* expression levels in the SCG of EGFP-MYCN⁺ larvae at 3 dpf (Fig 4D and 4H),

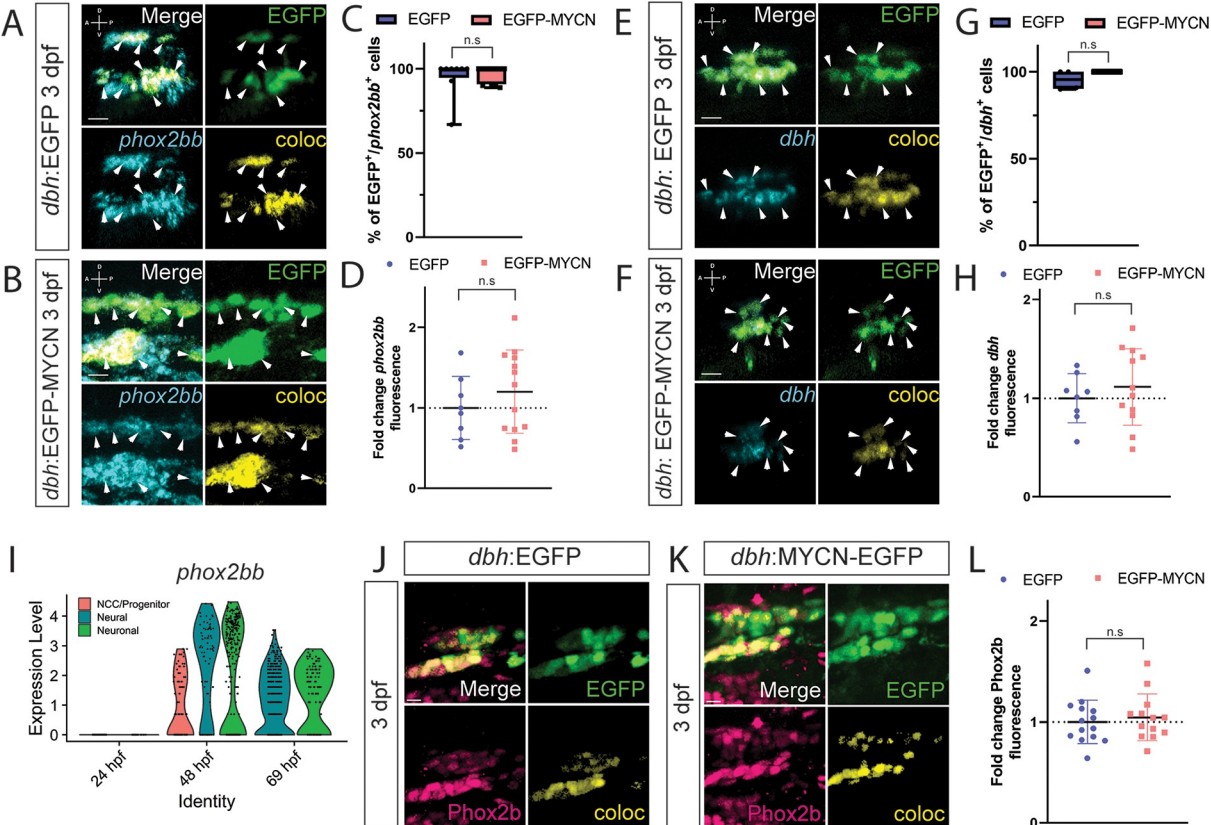

**Fig 4. MYCN overexpressing cells contain a SAP gene expression signature at 3 dpf.** A,B,E,F) WICHCR performed using HCR probes against *phox2bb* (A,B) or *dbh* (E,F) and with antibody against EGFP on 3 dpf *dbh*:EGFP (A,E) and *dbh*:EGFP-MYCN (E,F) larvae. Representative images reveal expression of the markers and their colocalized channel (coloc) within the SCG. C,G) Percentage of EGFP+ or EGFP-MYCN+ cells co-expressing *phox2bb* (C) or *dbh* (G) in 3 dpf larvae. For *phox2bb* EGFP n = 8, EGFP-MYCN n = 11; for *dbh* EGFP n = 4, EGFP-MYCN n = 4. D,H) Mean *phox2bb* (D) or *dbh* (H) fluorescence intensity quantified in the SCG and normalized to *dbh*:EGFP average intensity at 3 dpf. For *phox2bb* EGFP n = 8, EGFP-MYCN n = 12; for *dbh* EGFP n = 8, EGFP-MYCN n = 12. Markers: EGFP (green), *phox2bb* or *dbh* (cyan) and coloc channel (yellow). A (anterior), P (posterior), D (dorsal), V (ventral) axes shown in upper left corner. Scale bars = 10 μm. I) Violin plot showing *phox2bb* expression from single cell datasets of *sox10* derived cells at 24, 48, and 69 hpf. J,K) Representative images of SCG from *dbh*:EGFP (J) and *dbh*:EGFP-MYCN (K) larvae after immunofluorescence against Phox2b and EGFP at 3 dpf. L) Mean Phox2b fluorescence intensity quantified in the SCG and normalized to *dbh*:EGFP average intensity at 3 dpf. Markers: EGFP (green), Phox2b (pink). Scale bars = 7 μm. For all graphs n.s., non-significant (P>0.05).

EGFP-MYCN[+] larvae did not display statistically different levels of *phox2bb* (Fig 4D, p = 0.352334) or *dbh* (Fig 4H, p = 0.470920). These results are in agreement with the queried expression of *phox2bb*, *elavl3*, and *dbh* during normal NCC development in the single cell datasets, where we found these markers to be primarily expressed by neural and neuronal populations at later time points during their development (Fig 4I, S4 Fig). Since MYCN overexpression did not affect the mRNA expression of SAP differentiation markers, we performed immunohistochemistry against Phox2b, to assess if MYCN overexpression altered translation and/or protein localization of this SAP marker. We found that both EGFP[+] and EGFP-MYCN[+] larvae expressed Phox2b protein in their developing SCGs at 3 dpf (Fig 4J and 4K) with no significant changes in their levels (Fig 4L). When combined, these data indicate that MYCN overexpression does not affect the expression or localization of the SAP markers *phox2bb* and *dbh* within the SCG.

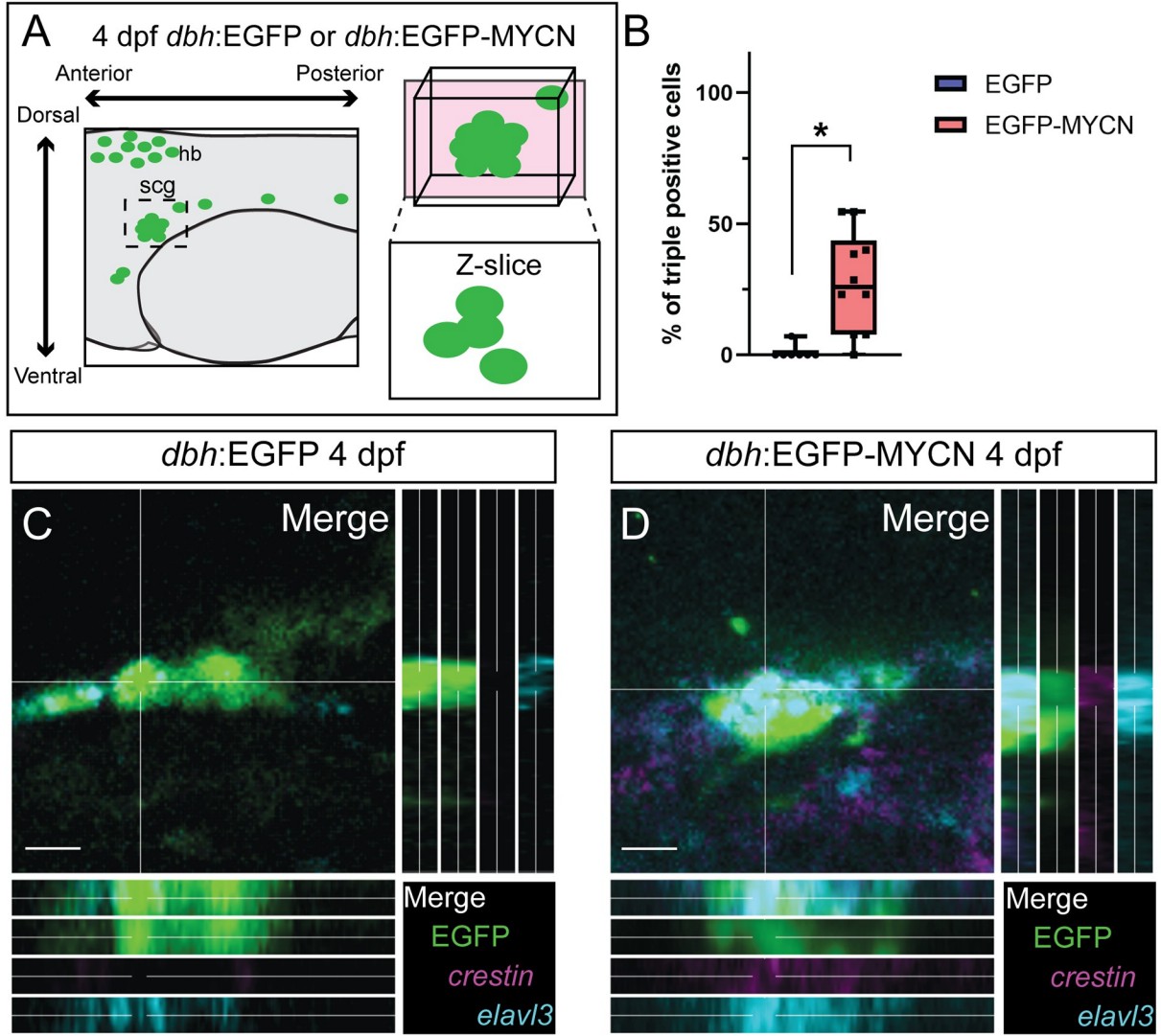

**Fig 5. Ectopic MYCN expression results in an aberrant SAP population that co-expresses NCC and neuronal markers at 4 dpf.** A) Cartoon depicting the z-slice through the SCG for quantifying individual triple positive (EGFP+/crestin+/elavl3+) cells. B) Percentage of triple positive cells (EGFP+/crestin+/elavl3+) detected in dbh:EGFP or dbh:EGFP-MYCN larvae at 4 dpf. For EGFP n = 7, EGFP-MYCN n = 10. C,D) Representative images from sections through the developing SCG in dbh:EGFP (C) or dbh:EGFP-MYCN (D) larvae at 4 dpf. WICHCR performed using HCR probes against crestin, elavl3, and with antibody against EGFP. Triple positive (EGFP+/crestin+/elavl3+) cells can be seen in MYCN overexpressing larvae. Markers: EGFP (green), crestin (magenta), and elavl3 (cyan). Scale bar = 10 μm. * P<0.05.

### Ectopic MYCN expression in SAP cells leads to an abnormal cellular population that co-expresses NCC and neuronal differentiation genes

Since SAP markers within the MYCN overexpressing cells were not lost, we next asked if EGFP-MYCN+ cells co-expressed NCC and neuronal differentiation markers. To that end, we assayed the expression of crestin, and elavl3, as markers for NCC and neuronal differentiation, respectively, at 4 dpf (Fig 5). Interestingly, we found that an average of 27.77% of cells within the SCG of EGFP-MYCN+ larvae presented simultaneous expression of both crestin and elavl3 markers (Fig 5B), whereas SAPs in control larvae presented only elavl3 (Fig 5B). By analyzing slice views of SCG confocal z-stacks, we were able to confirm that individual

EGFP-MYCN[+] cells presented both *crestin* and *elavl3* transcripts concurrently (Fig 5D), while EGFP[+] control cells did not present *crestin* (Fig 5C). Additionally, we explored whether any of these aberrant *crestin*[+]/*elavl3*[+] MYCN-overexpressing cells were proliferating by using phosphorylated Histone H3 (pHH3), a well-known marker for cell division [58]. In EGFP control larvae, we did not detect any *elavl3*[+]/pHH3[+] cells in the SCG at 4 dpf, and we only detected one EGFP-MYCN[+] larval fish that presented a *crestin*[+]/*elavl3*[+]/pHH3[+] cell (S5 Fig), suggesting that proliferative (pHH3[+]) aberrant (*crestin*[+]/*elavl3*[+]) cells were rare at the time point analyzed.

Taken together, data from Figs 2–5 show that, when compared with controls, EGFP-MYCN[+] cells display an ectopic transcriptomic expression signature that includes genes from NCC, SAP, and neuronal differentiation modules. These results suggest that MYCN overexpression is sufficient to alter SAP differentiation by promoting the expansion of NCC gene expression programs from 3 to 5 dpf, early timepoints during SAP development.

## MYCN overexpression alters BMP signaling during SAP differentiation

Given the role BMP plays in early SAP development [10, 29, 59], we aimed to characterize if MYCN overexpression in the SAP could alter BMP signaling dynamics. For this, we performed immunohistochemistry against pSmad1/5/8, as readout for BMP signaling activity [60], at 3 dpf and found that there was a decrease of pSmad1/5/8 signal within the SCG in EGFP-MYCN[+] fish (Fig 6B), compared to EGFP[+] control (Fig 6A). We quantified mean pSmad1/5/8 fluorescence intensity and found that EGFP-MYCN[+] larvae presented significantly lower levels of pSmad1/5/8 compared with control larvae (Fig 6C, p <0.000001). Furthermore, we used WICHCR to corroborate if BMP activity was downregulated by MYCN overexpression in the SAP lineage. Specifically, we assayed the expression of *id2a*, a BMP pathway target gene [61]. We found a significant reduction of *id2a* expression levels in the SCG region of MYCN overexpressing larvae when compared to EGFP control larvae (S6 Fig). Taken together, these results suggest ectopic MYCN overexpression dampens the BMP signaling pathway during SAP development.

We then asked if chemical inhibition of the BMP pathway could recapitulate the effect MYCN overexpression had regarding the aberrant expression of NCC genes in developing SCG of zebrafish larvae. For this, we used K02288 or Dorsomorphin, BMP inhibitors previously validated [62, 63], to attenuate BMP signaling at 48 hpf, a time during the onset of sympathetic differentiation [48]. After 24 hours of incubation with either DMSO, as control, or BMP inhibitor (K02288 or Dorsomorphin) we fixed the larvae and performed WICHCR to detect expression levels of *sox10* and *phox2bb*, for marking NCC and SAP cells, respectively (Fig 6D). Interestingly, we found that after treatment with K02288 or Dorsomorphin, EGFP[+] larvae presented expanded expression of *sox10*, similar to EGFP-MYCN[+] larvae after treatment in either DMSO, K02288, or Dorsomorphin conditions (Fig 6F and 6G). However, the expression of *phox2bb* persisted. To further explore if *sox10* was differentially expressed in the K02288 or Dorsomorphin-treated larvae, when compared with EGFP[+] larvae treated with DMSO, we measured mean *sox10* intensity in the developing SCG at 3 dpf (Fig 6E). After treatment with the BMP inhibitors, EGFP[+] larvae displayed significantly higher levels of *sox10* in K02288 (Fig 6E, p = 0.000830) and Dorsomorphin (Fig 6E, p = 0.0013) conditions, when compared with DMSO. Equally, EGFP-MYCN[+] larvae retained higher expression of *sox10* regardless of DMSO (Fig 6E, p = 0.025682), K02288 (Fig 6E, p = 0.002961), or Dorsomorphin (Fig 6E, p = 0.0171) treatment, when compared with EGFP[+] larvae in DMSO. To further corroborate if BMP inhibition could be leading to the expression of an aberrant NCC signature, we assayed *crestin* and *elavl3* as markers of NCC and SAP differentiation, respectively. We

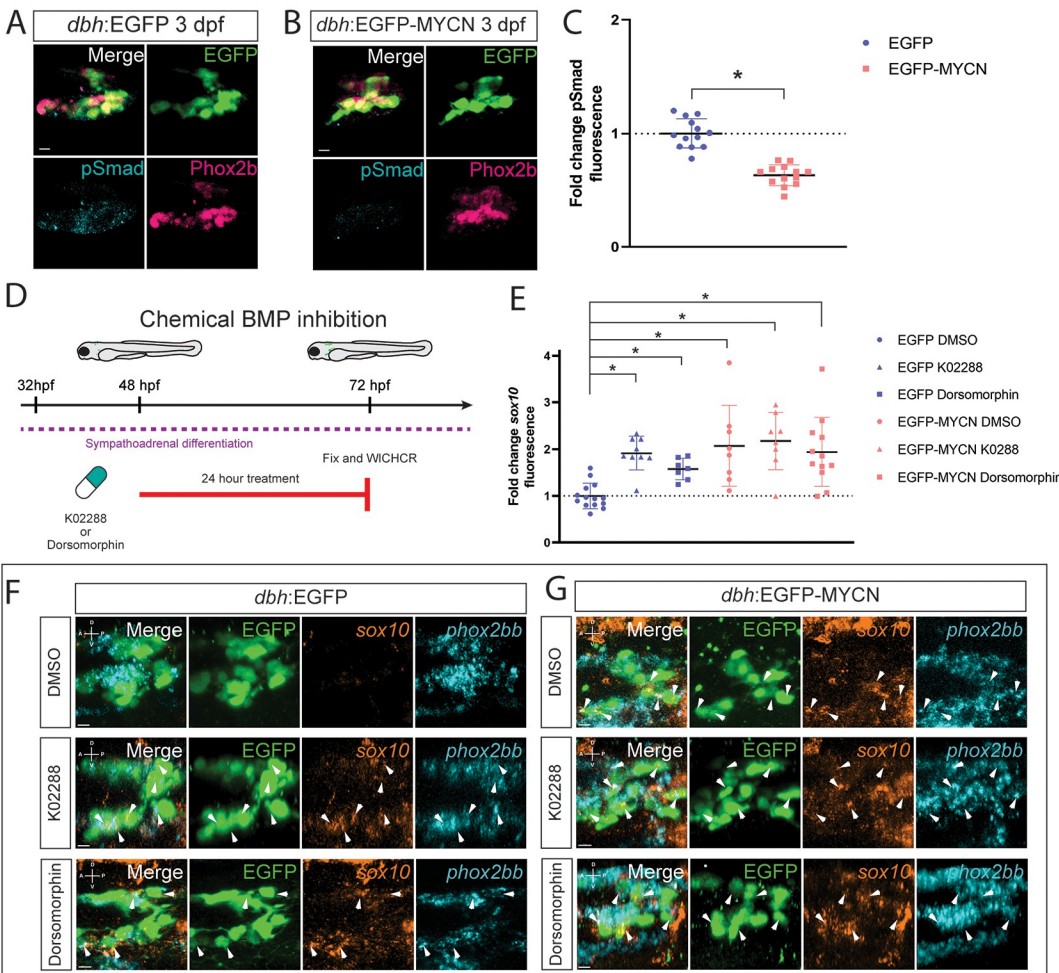

**Fig 6. BMP signaling activity is dampened within developing EGCP-MYCN+ larvae.** A,B) Representative images of SCG from *dbh*:EGFP (A) and EGFP-MYCN (B) larvae after immunofluorescence against Phox2b and EGFP. Markers: EGFP (green), pSmad1/5/8 (cyan), and Phox2b (pink). C) Mean pSmad1/5/8 fluorescence intensity quantified in the SCG and normalized to *dbh*:EGFP average intensity at 3 dpf. For EGFP n = 13, EGFP-MYCN n = 13. D) Schematic of treatment with 10μM K02288 or 50 μM Dorsomorphin embryos were treated at 48 hpf for 24 hours and then fixed at 72 hpf. E) Mean *sox10* fluorescence intensity quantified in the SCG and normalized to *dbh*:EGFP average intensity at 3 dpf. For EGFP DMSO n = 14, EGFP K02288 n = 9, EGFP Dorsomorphin n = 7, EGFP-MYCN DMSO n = 8, EGFP-MYCN K02288 n = 8, EGFP-MYCN Dorsomorphin n = 12. F,G) Representative images from developing SCG in *dbh*:EGFP (F) or *dbh*:EGFP-MYCN (G) larvae at 3 dpf after 24 h treatment with either DMSO (F,G, upper panels),K02288 (F,G, middle panels), or Dorsomorphin (F,G, lower panels). WICHCR performed using HCR probes against *sox10*, *phox2bb*, and with antibody against EGFP. K02288 and Dorsomorphin treatments cause expansion of *sox10* expression in EGFP⁺ larvae, similar to the expansion seen in MYCN⁺ larvae in either condition. Markers: EGFP (green), *sox10* (orange), and *phox2bb* (cyan). Scale bar = 7 μm. For all graphs * P<0.05, n.s., non-significant (P>0.05).

found that, after treatment with either K02288 or Dorsomorphin, EGFP⁺ larvae presented a slight increase in *crestin* expression (S7 Fig), while *elavl3* remained virtually unchanged (S7 Fig). These results shed light on a potential mechanism behind MYCN overexpression and its effect on correct SAP differentiation by modulation of BMP signaling. We propose a model where MYCN overexpression disrupts the normal NCC to SAP neuronal cell differentiation transition, possibly via alterations in BMP signaling (Fig 7). Cells with controlled *mycn* expression over time are able to respond to BMP signals and reach their final fate (Fig 7B). On the

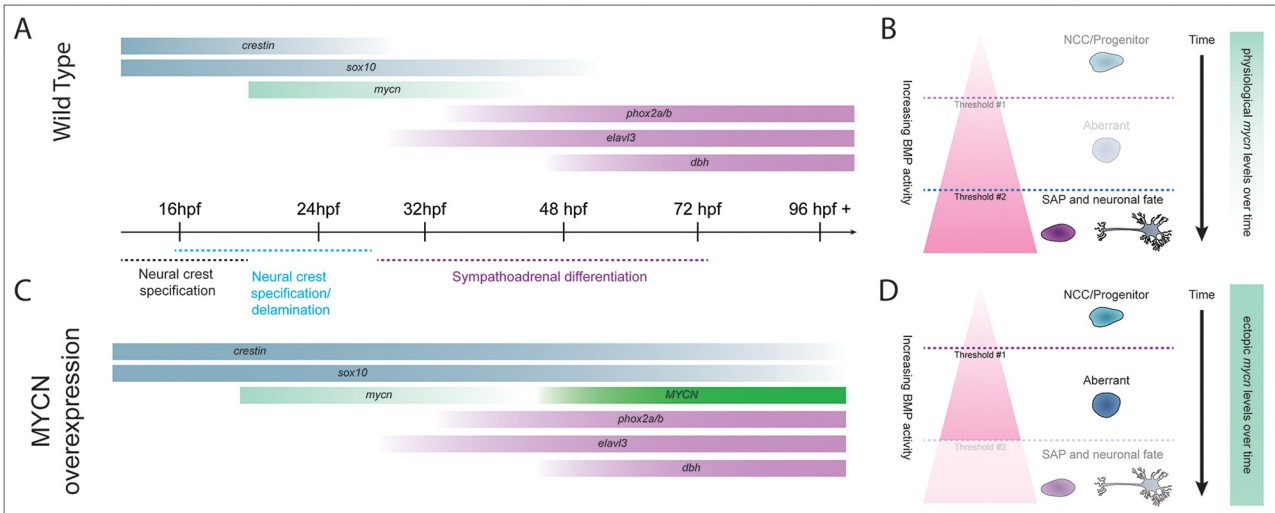

**Fig 7. MYCN overexpression produces a cellular population with aberrant gene expression and a dampened BMP response.** A,C) Normal NCC differentiation towards SAP fate requires a fine-tuned expression of transcription factors and differentiation effectors. *sox10* and *crestin* (in zebrafish) are expressed during early NCC specification and retain their expression during NCC migration. As NCC undergo epithelial to mesenchymal transition (EMT) and migrate they transiently express *mycn*. When NCC receive SAP differentiation signals, they downregulate expression of early NCC genes and commence the expression of genes required for sympathoadrenal specification like *phox2bb*, *elavl3* and *dbh*. C) The ectopic expression of MYCN causes a change in the developmental program and timing of these cells, where NCC markers like *crestin* and *sox10*, present expanded expression concurrent with SAP differentiation markers like *phox2bb*, *elavl3*, and *dbh*. B,D) Proposed model where MYCN overexpression disrupts SAP neuronal differentiation, possibly via alterations in BMP signaling. B) SAP development progresses correctly when the physiological levels of MYCN are controlled over time, and cells are able to respond to BMP signals and reach their final fate. D) In MYCN-overexpressing conditions, cells can no longer establish a proper BMP response to undergo neuronal differentiation, creating an aberrant population that contains NCC and SAP markers.

other hand, cells with MYCN overexpression fail to establish a BMP response strong enough to finalize differentiation and are instead stuck in an aberrant state where they express NCC and SAP markers (Fig 7D).

## Discussion

During development and organogenesis, the precise execution of cell fate specification programs ensures the timely differentiation of cells and tissues. In this study, we found that when ectopic MYCN was autonomously expressed during the earliest stages of SAP lineage development in zebrafish, the presence of a transient, aberrant gene expression profile in which NCC, SAP, and neuronal differentiation markers are co-expressed was detected. We imaged the *in vivo* presence of this transient expansion, revealing a critical window, namely from 3 to 5 dpf, during which MYCN alters NCC marker activation. We also discovered that MYCN overexpression led to a dampening of BMP signaling response in the SAP lineage, as measured by reduced presence of phosphorylated-Smad, and diminished expression of BMP target *id2a*. Furthermore, pharmacological inhibition of the BMP pathway alone was sufficient to induce an aberrant NCC signature in SAP cells, phenocopying ectopic MYCN.

Given that transient expression of MYCN is necessary for correct migration and differentiation of NCC [45], but amplification and/or high levels of MYCN expression are linked to high-risk tumors [33, 37, 64, 65], it is important to investigate MYCN's role during early SAP development *in vivo*. While the exact developmental origin of NB is still debated, it has generally been thought to arise from the NC-derived sympathoadrenal lineage [66]. Various recent studies have described the transcriptional signatures of sympathoadrenal cells and NB types, as well as NB's ability to remain undifferentiated, highlighting NB similarities to undifferentiated

sympathoadrenal progenitor states during development [67–70] In addition, a previous study explored the effect of ectopic MYCN overexpression in isolated murine SAPs [71], where they showed the presence of sympathetic lineage markers, *Dbh*, *Th*, and *Phox2b*. Furthermore, a study using a TH-MYCN mouse model showed that post-natal Phox2B[+] progenitors were arrested in a proliferative progenitor state where overt neuronal differentiation did not occur [72].

Our results agree with the prior data that SAP markers are present in MYCN overexpressing cells; however, our work expands temporal information regarding the transcriptional identity of these cells by unveiling the transient expansion, from 3 to 5 dpf, of an aberrant NCC signature at the onset of SAP development, but that also paradoxically co-expresses neuronal differentiation genes. Indeed, we showed that EGFP-MYCN[+] cells presented simultaneous expression of *crestin*, *sox10*, and *elavl3* markers, while EGFP[+] control cells only expressed *elavl3*. We further explored these findings with *in vivo* imaging using a *sox10* reporter line [55] and found a transient expansion of *sox10* reporter activity in the SCG of MYCN overexpressing larvae. Sox10 has been shown to induce the expression of Phox2b, but sustained overexpression of Sox10 is able to block overt neuronal differentiation [27]. Taken together, these results suggest a spatiotemporal role for MYCN during the cessation of NCC programs and correct sympathetic development, where if MYCN is constitutively present, it affects the normal expression dynamics of critical differentiation effectors, like Sox10. This atypical phenotype could potentially be leading MYCN overexpressing cells to not properly differentiate, thus promoting a poorly differentiated population that could prime the organism for disease onset.

NCCs require a conjunction of signaling pathways to be activated at certain differentiation states. The fine-tuning of such pathways may be required for NCC to acquire SAP fate, maintain the progenitor in an undifferentiated state, or allow the neural progenitor to continue its fate towards a mature neural sympathetic cell. Given that SAPs receive cues from the surrounding tissue microenvironment, that directs their migration and final cell fates, it is critical to study the effects MYCN overexpression has on SAP differentiation. BMPs induce sympathetic differentiation of NCCs *in vitro* and *in vivo* [29–31, 73], whereby it elicits the expression of multiple key SAP specification transcription factors like Hand2, Gata3, Ascl1a, and Phox2a/Phox2b, which together promote further differentiation while enhancing proliferation and survival of the sympathoadrenal lineage [10, 12]. A link between BMP signaling and MYCN has been reported in the context of NCC to neuronal differentiation *in vitro*, where MYCN is downregulated after treatment with BMP-4 [74]. However, the effect of MYCN overexpression on the BMP response during SAP maturation has not been dissected.

In this work, we discovered that MYCN overexpression alters BMP pathway activity during early sympathetic development. After analyzing BMP activity via pSmad1/5/8 and *id2a* target gene expression, we found that MYCN overexpression attenuated BMP response in the SCG of developing zebrafish larvae. What's more, is that 24-hour chemical attenuation of the BMP pathway during early SAP development expanded the expression of NCC marker *sox10 in vivo*. These data phenocopied the effect MYCN overexpression had on the concurrent expression of NCC and SAP markers (Figs 2–5). Taken together, these results suggest MYCN could be promoting an aberrant NCC-like state and a dampening of the BMP signaling response. Overall, our data point to a model where MYCN overexpression produces a cellular population with aberrant gene expression, where both NCC and SAP differentiation genes are co-expressed, that fails to properly acquire their final SAP fate (Fig 7). Specifically, the MYCN overexpressing population presents a transient increase in *crestin* and *sox10* up to 5 dpf (Fig 7C), when compared to wildtype SAP development where the expression of these genes is mostly gone by 3 dpf (Fig 7A). As well, this aberrant gene expression is associated with a

dampening of the BMP response (Fig 7B and 7D). We postulate a model where in wildtype conditions, physiological levels of *mycn* are controlled over time, and NCCs can establish an appropriate response to BMP signaling, thus undergoing correct SAP developmental timing, and overt neuronal maturation (Fig 7B). On the other hand, when *mycn* levels are constant and elevated, cells are not able to maintain a correct BMP response to properly undergo final neuronal differentiation, and instead remain in an aberrant state where NCC and SAP genes are co-expressed (Fig 7D).

Overall, this work increases our understanding of MYCN overexpression effects in NCC and sympathetic development by identifying a mechanism by which MYCN alters the normal NCC to SAP differentiation programs, leading to the aberrant population during early zebrafish development. We have also discovered that MYCN overexpression leads to a dampening in BMP response in the SCG, shedding light on the mechanisms required for correct SAP differentiation. In the very early stages of SAP development, this ultimately creates an aberrant, transient population that contains NCC and SAP markers, which may have implications for further priming of a neoplastic lesion in the future.

## Methods

### Animal husbandry and tissue collection

All experiments in this work were carried out in accordance with the guidelines of the Rice University Institutional Animal Care and Use Committee (protocol 1143754). Adult zebrafish were carefully bred to ensure synchronous embryo collection. The following lines were used for the experiments: AB WT, *dbh*:EGFP [43], *dbh*;EGFP-MYCN [43], *sox10*:mRFP [55]. After collection, the embryos were maintained in standard E3 media until 24 hpf and then kept in 0.003% 1-phenyl 2-thiourea (PTU)/E3 solution [75], to prevent melanin formation, until fixation at the stages noted in each experiment. Embryos were sorted for fluorescence at 48 hpf and kept continuous monitoring of their fluorescence. Embryos and larvae that presented developmental delay or defects were removed prior to fixation.

### Single cell data set analysis

Publicly accessible RDS files from single cell RNAseq datasets of NCC and NCC-derived cells from 24 [50], 48–50 and 68–70 hpf zebrafish embryos [49, 50] (GEO accession: GSE152906 and GSE163907) were analyzed as previously described [49] using Seurat [76–78] v5.0.1 for R. Previously identified, based on marker expression as described before [49, 50], neural crest, neuronal, and neural cell type populations were subsetted. Target gene expression was assayed on these selected populations at 24, 48–50, 68–70 hpf during early NCC development and SAP differentiation, respectively. Violin plots for data visualization were generated using the VlnPlot command.

### Whole mount immunofluorescence and Whole-Mount Immuno-Coupled Hybridization Chain Reaction (WICHCR)

HCR probes against zebrafish *crestin* (AF195881.1), *sox10* (AF195881.1), *phox2bb* (NM_001014818.1), *elavl3* (NM_131449), *dbh* (NM_001109694.2), and *id2a* (NM_201291.1) transcripts were generated and purchased from Molecular Instruments, Inc. Whole Mount Immuno-Coupled HCR (WICHCR) experiments were performed as described previously [54]. Whole mount immunohistochemistry was executed according to methods previously described [79]. To detect protein expression, the following antibodies and concentrations were used: rabbit polyclonal IgG anti-GFP (Invitrogen Molecular Probes, A11122, 1:500), rabbit

polyclonal anti-p-Smad1/5/9 IgG (Cellular Signaling Technologies, 13820S, 1:500), and mouse monoclonal IgG1 anti-Phox2b [80] (Santa Cruz Biotechnology, SC-376997, 1:250). Secondary antibodies and concentrations used were Alexa Fluor 488 goat anti-rabbit IgG (H+L) (Invitrogen, A11008, 1:500), and Alexa Fluor 594 goat anti-mouse IgG1 (Invitrogen, A21125, 1:500). Prior to imaging, embryos were cleared in serial glycerol dilutions and mounted in 75% glycerol/ 25% PBST for imaging in confocal microscope, to achieve a high-resolution image. Samples were imaged using an Olympus FV3000 Laser Scanning Confocal, with a long working distance 20.0× objective (UCPLFLN20X/0.70NA) objective. Final images were exported for analysis in IMARIS image analysis software V9.8.2 (Bitplane).

## Image processing and analysis on IMARIS software

**Region of interest generation, cell detection and quantification.** Using IMARIS image analysis software V9.8.2, the Crop 3D function was used to generate regions of interest (ROI) within the developing superior cervical ganglion (SCG). The dimensions for the ROI generated were 60 X 60 X 60 um, creating a cube centering the SCG (S1 Video). For cell identification, the "spots" function was used to identify the number of cells (spots) with a 10 μm diameter (average SAP cell diameter in zebrafish) positive for EGFP (S1 Video). Manual curation and validation of EGFP$^+$ and EGFP-MYCN$^+$ cells were carried out after unbiased spots detection. Number of cells (spots) were compared across the conditions and timepoints.

**Colocalization channel generation.** The "coloc" function in IMARIS was utilized. Briefly, a threshold was selected for each of the channels analyzed. Then, IMARIS detected pixels that presented signal in both fluorescence channels higher than the threshold parameters set and marked them as a new coloc channel. For our analyses, thresholds were set using images from EGFP control larvae for the individual genes of interest (GOI) analyzed. These same thresholds (made for each GOI) were then applied to their respective EGFP-MYCN overexpressing larvae counterparts.

**Transcript expression and double/triple positive cell identification.** After individual EGFP$^+$ or EGPF-MYCN$^+$ cells were identified, additional background "spots" were detected to identify the basal fluorescence signal in the GOI channel. A threshold was set by averaging the mean intensity fluorescence detected in the background spots. Later, double and triple positive cells were identified by detecting if an EGFP$^+$ or EGFP-MYCN$^+$ "spot" had higher mean intensity fluorescence than the background "spots" average and counted as a positive cell for that particular GOI.

**Surface analysis.** Using the surface function in IMARIS, complete SCG area was detected. The surface was manually generated by tracing the outline of the SCG for each z-plane throughout the stack, taking good care to include the entirety of the volume occupied by EGFP signal in the region analyzed. Mean fluorescence intensity was calculated for the surface to compare fluorescence levels across conditions.

## Live confocal time-lapse microscopy

Embryos were anesthetized using 0.4% Tricaine (Sigma-Aldrich, A5040), then mounted in 15 μ-Slide 4 Well imaging chambers (Ibidi, 80427) using 1.0% low melt temperature agarose dissolved in E3 media. Embedded embryos were then covered in 1× PTU/E3 media supplemented with 0.4% Tricaine. Larvae were imaged, at a constant temperature of 28˚C maintained by OKOLAB Uno-controller imaging incubator, using an Olympus FV3000 Laser Scanning Confocal, with a long working distance 20.0× objective (UCPLFLN20X/0.70NA) objective. Final time lapse images were exported for analysis in IMARIS image analysis software (Bitplane). Region of interest generation and coloc analysis was performed for each

timepoint during the timelapse as explained above. Care was taken to ensure SCG was always maintained within the center of the ROI generated.

## Chemical inhibitor treatment

BMP inhibitors, K02288 (SML1307, Sigma Aldrich) [63] or Dorsomorphin (P5499, Sigma Aldrich) [63], master stocks were diluted in DMSO, working stocks were generated by diluting the master stock in 1xPTU/E3 medium to achieve a 10 micromolar concentration for K02288 and 50 micromolar concentration for Dorsomorphin. 10 embryos per well were incubated in 1.5 mL of diluted DMSO, K02288, or Dorsomorphin in 1xPTU/E3 medium. Drug incubation started at 48 hpf and lasted for 24 hours, after which the larvae were collected at 72 hpf. Larvae were processed and imaged as described above.

## Graphical representation and statistics

Spots counts and mean fluorescence intensity for spots and surfaces data were exported from IMARIS, curated and analyzed for graphical representation using GraphPad Prism (version 9.5.1). Statistical analysis was performed in GraphPad Prism (version 9.5.1), using two-tailed unpaired t-test. For all graphs, * $P<0.05$, n.s., non-significant ($P>0.05$).

## Supporting information

**S1 Fig. *mycn* expression is largely restricted to early time points during zebrafish development.** A) Violin plot showing expression of *mycn* is restricted to progenitor and NCC populations during early timepoints. B,C) representative images of whole mount in situ hybridization against *mycn* at 48 hpf (B) and 72 hpf (B) Anterior is shown to the left. A (anterior), P (posterior), D (dorsal), V (ventral) axes shown in lower right corner. b = developing brain, ba = developing branchial arches, g = developing gut, e = developing eye.
(TIF)

**S2 Fig. MYCN overexpression expands *crestin* and *sox10* expression *in vivo*.** A,B,C,D) representative confocal images of SCG from *dbh*:EGFP (A,B) and *dbh*:EGFP-MYCN (C,D) larvae with WICHCR against *crestin* and EGFP. E,F,G,H) representative confocal images of SCG from *dbh*:EGFP (E,F) and *dbh*:EGFP-MYCN (G,H) larvae with WICHCR against *sox10* and EGFP. Markers: EGFP (green), *crestin* (magenta), and coloc channel (yellow). A (anterior), P (posterior), D (dorsal), V (ventral) axes shown in upper left corner. Scale bars = 10 μm. I-L) Percentage of EGFP⁺ or EGFP-MYCN⁺ cells that also express *crestin* (I,J) or *sox10* (K,L) at 4 dpf (I,K), and 5 dpf (J,L). For *crestin* 4 dpf EGFP n = 14, EGFP-MYCN n = 14; for *crestin* 5 dpf EGFP n = 12, EGFP-MYCN n = 10. For *sox10* 4 dpf EGFP n = 11, EGFP-MYCN n = 10; for *sox10* 5 dpf EGFP n = 9, EGFP-MYCN n = 10. For all graphs * denotes $P<0.05$.
(TIF)

**S3 Fig. MYCN overexpression does not affect *phox2bb* expression in the SCG at 3 dpf.** A,B, D,E) representative confocal images of SCG from *dbh*:EGFP (A,D) and *dbh*:EGFP-MYCN (B, E) larvae with WICHCR against *phox2bb* and EGFP. Markers: EGFP (green), *phox2bb* (cyan), and coloc channel (yellow). A (anterior), P (posterior), D (dorsal), V (ventral) axes shown in upper left corner. Scale bars = 10 μm. C,F) Percentage of EGFP⁺ or EGFP-MYCN⁺ cells that also express *phox2bb* at 4 dpf (C) and 5 dpf (F). For 4 dpf EGFP n = 9, EGFP-MYCN n = 9; for 5 dpf EGFP n = 9, EGFP-MYCN n = 10. For all graphs n.s. denotes non-significant ($P>0.05$).
(TIF)

**S4 Fig. Expression of *dbh* and *elavl3* during early zebrafish development of the *sox10* lineage.** A,B) Violin plots depicting *dbh* (A) and *elavl3* (B) expression from single cell datasets of *sox10*-derived cells at 24, 48, and 69 hpf, as described in methods.
(TIF)

**S5 Fig. Aberrant cells that express NCC, SAP, and proliferation markers concurrently are rare.** A,B) Images from sections through the developing SCG in *dbh*:EGFP (A) or *dbh*:EGFP-MYCN (B) larvae at 4 dpf. WICHCR performed using probes against *crestin*, *elavl3*, and with antibodies against EGFP and phosphorylated Histone H3 (pHH3). Proliferating SAP cells (pHH3[+]) were not detected in the SCG of either conditions. Only one MYCN-overexpressing larvae presented a proliferating (pHH3[+]/*crestin*[+]/*elavl3*[+]) cell. Markers: EGFP (green), *crestin* (magenta), *elavl3* (cyan), and pHH3 (yellow). Scale bar = 10 μm.
(TIF)

**S6 Fig. MYCN overexpression leads to decreased *id2a* expression in early SCG.** A,B) Confocal images of the developing SCG at 3 dpf in *dbh*:EGFP (A) or *dbh*:EGFP-MYCN (B) larvae. Markers: EGFP (green), *id2a* (cyan). A (anterior), P (posterior), D (dorsal), V (ventral) axes shown in upper left corner. Scale bars = 7 μm. C) Mean *id2a* fluorescence intensity quantified in the SCG and normalized to *dbh*:EGFP average intensity at 3 dpf. For EGFP n = 12, EGFP-MYCN n = 11. * P<0.05.
(TIF)

**S7 Fig. Chemical inhibition of BMP signaling leads to ectopic *crestin* expression in zebrafish SCG at 3 dpf.** A,B) Representative images from developing SCG in *dbh*:EGFP (F) or *dbh*:EGFP-MYCN (G) larvae at 3 dpf after 24 h treatment with either DMSO (A,B, upper panels), K02288 (A,B, middle panels), or Dorsomorphin (A,B, lower panels). WICHCR performed using HCR probes against *crestin*, *elavl3*, and with antibody against EGFP. K02288 and Dorsomorphin treatments cause a discrete expansion of *crestin* expression in EGFP[+] larvae. C,D) Mean *crestin* (C) and *elavl3* (D) fluorescence intensity quantified in the SCG and normalized to *dbh*:EGFP average intensity at 3 dpf. For EGFP DMSO n = 11, EGFP K02288 n = 9, EGFP Dorsomorphin n = 9, EGFP-MYCN DMSO n = 10, EGFP-MYCN K02288 n = 9, EGFP-MYCN Dorsomorphin n = 8. F,G) Markers: EGFP (green), *crestin* (magenta), and *elavl3* (cyan). Scale bar = 7 μm. For all graphs * P<0.05, n.s., non-significant (P>0.05).
(TIF)

**S1 Movie. Animated 360˚ view of ROI generation and spots detection on IMARIS software.** SCG was identified, marked as ROI, and isolated for further spots detection and manual curation of EGFP[+] cells using IMARIS software. Markers: EGFP (green), *crestin* (magenta).
(MP4)

**S2 Movie. *in vivo* time-lapse of control larvae from 70 hpf to 118 hpf.** Control larvae *dbh*:EGFP crossed with the *sox10* reporter *sox10*:mRFP line. EGFP and mRFP channels are shown. Markers: EGFP (green), mRFP (magenta).
(MP4)

**S3 Movie. Merged channel movie of *in vivo* time-lapse of SCG in control larvae from 70 hpf to 118 hpf.** SCG of control larvae *dbh*:EGFP crossed with the *sox10* reporter *sox10*:mRFP line. SCG was detected and isolated from S2 Movie. Merged channels are shown. Markers: EGFP (green), mRFP (magenta), coloc (yellow).
(MP4)

**S4 Movie. EGFP channel movie of *in vivo* time-lapse of SCG in control larvae from 70 hpf to 118 hpf.** SCG of control larvae *dbh*:EGFP crossed with the *sox10* reporter *sox10*:mRFP line. SCG was detected and isolated from S2 Movie. EGFP channel is shown. Markers: EGFP (green).
(MP4)

**S5 Movie. mRFP channel of *in vivo* time-lapse of SCG in control larvae from 70 hpf to 118 hpf.** SCG of control larvae *dbh*:EGFP crossed with the *sox10* reporter *sox10*:mRFP line. SCG was detected and isolated from S2 Movie. mRFP channel is shown. Markers: mRFP (magenta).
(MP4)

**S6 Movie. Coloc channel of *in vivo* time-lapse of SCG in control larvae from 70 hpf to 118 hpf.** SCG of control larvae *dbh*:EGFP crossed with the *sox10* reporter *sox10*:mRFP line. SCG was detected and isolated from S2 Movie. coloc channel is shown. Markers: coloc (yellow).
(MP4)

**S7 Movie. Merged channel movie of *in vivo* time-lapse of MYCN overexpressing larvae from 70 hpf to 118 hpf.** MYCN overexpressing larvae *dbh*:EGFP-MYCN crossed with the *sox10* reporter *sox10*:mRFP line. Merged channels are shown. Markers: EGFP (green), mRFP (magenta).
(MP4)

**S8 Movie. Merged channel movie of *in vivo* time-lapse of SCG in MYCN overexpressing larvae from 70 hpf to 118 hpf.** SCG of MYCN overexpressing larvae *dbh*:EGFP-MYCN crossed with the *sox10* reporter *sox10*:mRFP line. SCG was detected and isolated from S7 Movie. Merged channels are shown. Markers: EGFP (green), mRFP (magenta), coloc (yellow).
(MP4)

**S9 Movie. EGFP channel movie of *in vivo* time-lapse of SCG in MYCN overexpressing larvae from 70 hpf to 118 hpf.** SCG of MYCN overexpressing larvae *dbh*:EGFP-MYCN crossed with the *sox10* reporter *sox10*:mRFP line. SCG was detected and isolated from S7 Movie. EGFP channel is shown. Markers: EGFP (green).
(MP4)

**S10 Movie. mRFP channel movie of *in vivo* time-lapse of SCG in MYCN overexpressing larvae from 70 hpf to 118 hpf.** SCG of MYCN overexpressing larvae *dbh*:EGFP-MYCN crossed with the *sox10* reporter *sox10*:mRFP line. SCG was detected and isolated from S7 Movie. mRFP channel is shown. Markers: mRFP (magenta).
(MP4)

**S11 Movie. Coloc channel movie of *in vivo* time-lapse of SCG in MYCN overexpressing larvae from 70 hpf to 118 hpf.** SCG of MYCN overexpressing larvae *dbh*:EGFP-MYCN crossed with the *sox10* reporter *sox10*:mRFP line. SCG was detected and isolated from S7 Movie. coloc channel is shown. Markers: coloc (yellow).
(MP4)

## Acknowledgments

We express our sincere gratitude to the entire Uribe Lab at Rice University for their insights and support for the completion of this work. We thank Dr. Budi Utama and the Rice University Shared Equipment Authority (SEA) on IMARIS image analysis suite. We also thank Hans F. Zimmer and John Powell for technical assistance.

## Author Contributions

**Conceptualization:** Rodrigo Ibarra-García-Padilla, Rosa A. Uribe.

**Data curation:** Rodrigo Ibarra-García-Padilla.

**Formal analysis:** Rodrigo Ibarra-García-Padilla, Rosa A. Uribe.

**Funding acquisition:** Rosa A. Uribe.

**Investigation:** Rodrigo Ibarra-García-Padilla, Annika Nambiar, Thomas A. Hamre, Eileen W. Singleton, Rosa A. Uribe.

**Methodology:** Rodrigo Ibarra-García-Padilla, Rosa A. Uribe.

**Project administration:** Rosa A. Uribe.

**Resources:** Rosa A. Uribe.

**Supervision:** Rosa A. Uribe.

**Validation:** Rodrigo Ibarra-García-Padilla, Annika Nambiar, Thomas A. Hamre, Eileen W. Singleton, Rosa A. Uribe.

**Visualization:** Rodrigo Ibarra-García-Padilla, Rosa A. Uribe.

**Writing – original draft:** Rodrigo Ibarra-García-Padilla, Rosa A. Uribe.

**Writing – review & editing:** Rodrigo Ibarra-García-Padilla, Annika Nambiar, Thomas A. Hamre, Eileen W. Singleton, Rosa A. Uribe.

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
