## [Decision Letter · Decision Letter 0]

18 Jun 2024

PONE-D-24-13321Expansion of a neural crest gene signature following ectopic MYCN expression in sympathoadrenal lineage cells in vivoPLOS ONE

Dear Dr. Uribe,

Thank you for submitting your manuscript to PLOS ONE. After careful consideration, we feel that it has merit but does not fully meet PLOS ONE’s publication criteria as it currently stands. Therefore, we invite you to submit a revised version of the manuscript that addresses the points raised during the review process.

Please address the reviewers' concerns point by point.  

We look forward to receiving your revised manuscript.

Kind regards,

Michael Klymkowsky, Ph.D.

Academic Editor

PLOS ONE

Journal Requirements:

Reviewers' comments:

Reviewer's Responses to Questions

**Comments to the Author**

1. Is the manuscript technically sound, and do the data support the conclusions?

Reviewer #1: No

Reviewer #2: Partly

2. Has the statistical analysis been performed appropriately and rigorously? 

Reviewer #1: Yes

Reviewer #2: Yes

3. Have the authors made all data underlying the findings in their manuscript fully available?

Reviewer #1: Yes

Reviewer #2: Yes

4. Is the manuscript presented in an intelligible fashion and written in standard English?

Reviewer #1: Yes

Reviewer #2: Yes

5. Review Comments to the Author

Reviewer #1: Interesting paper that analyzes effects of MYCN expression on SA lineage development in zebrafish in vivo. Authors note that MYCN blocks differentiation, demonstrate in a candidate gene approach that BMP signaling is downregulated in MYCN cells, and then phenocopy aspects of differentiation block using a BMP inhibitor.

1. Authors describe markers in neural crest development in the introduction, but fail to mention Crestin, which is the main marker they study early in the paper.

2. In Fig 5, are the crestin elavl3 double positive cells and the elavl3 control cells proliferating?

3. Can a structurally distinct inhibitor be used in addition to K02288 to assure the effects of this agent were on target? A rescue or genetic approach could also be used to achieve this.

4. Data on effects of BMPi are fairly preliminary, only 2 markers analyzed.

Reviewer #2: In this manuscript, Padilla et al. propose that MYCN overexpression early in SAP development leads to abnormal gene expression of NCC, SAP, and neuronal markers. The authors observe this transient change through live imaging, suggesting a critical period of time during which MYCN affects NCC marker activation, and they evaluate crosstalk between MYCN overexpression and BMP signaling by measuring levels of phosphorylated-Smad and the BMP target id2a. Finally, they use publicly available scRNAseq datasets of NCC and NCC-derived cells to strengthen their findings.

Major Points:

1. One overarching concern is that the focus on spatiotemporal changes is not well-defined (timepoints or range of time) for much of the presented data and conclusions. Thus, the overall scientific advance, particularly compared to previously published work, is difficult to understand and interpret in context. Examples include:

a) Line 198: When does SAP differentiation begin and how is that defined? Please explain what is meant by (hpf) “very early SAP development”.

b) Lines 98-101: “…it is known that MYCN expression is important for early NCC development… the role earlier ectopic MYCN overexpression has on NCC and SAP development has not been fully characterized…” While not necessarily contrary statements, this is confusing as written. Again, timepoints and a clearer statement about what the authors are referring to might help.

c) Lines 238-240: “…MYCN overexpression is sufficient to severely alter the NCC to SAP cell differentiation gene expression program, across space and time, during early SAP development...” ‘Severely altering’ and ‘across space and time’ overstate conclusions and also appear to be at odds with the authors’ own ideas, given preservation of parts of the gene expression program and the emphasis throughout on specific time ranges (not across time), respectively.

d) Figure 3 uses 70 hpf embryos. If SCG development starts around 48 hpf, please clarify why 70 hpf was the focus here.

e) An in depth explanation of the previously known timing and regulation of MYCN's impact on SAP differentiation would be generally helpful.

f) In Figure 4 (A-H), please justify/contextualize use of the 3 dpf timepoint.

g) Line 279: “…critical window…” Please define what the authors believe that to be.

h) Lines 296-299: “…our work expands spatiotemporal information regarding the transcriptional identity of these cells by unveiling the transient expansion…” Same request.

2. The abstract states “dampened BMP signaling activity…suggesting BMP is functionally important for the NCC to SAP differentiation transition.” Disagree. Please modify this overstatement.

3. Lines 221-223: It appears that “does affect” should read “does NOT affect”.

4. Lines 289-291: “Defective NCC specification towards SAP differentiation has been suggested as the origin mechanism for NB, where it is hypothesized that defects in cell fate acquisition promote a cell population that can form neoplastic lesions and give rise to NB (24,31).” There are several overlapping concerns here: Two reviews 5-6 years ago are used to justify this statement. Neither review appears to make this point, and there may be some field-specific confusion about terminology such has ‘origins’ and ‘derived from’. While NB is accepted as a ‘NC-derived cancer’, that is viewed by most of the field to mean origination from NC-derived sympathoadrenal progenitors/differentiated cells, not direct NC derivation as the authors appear to state (first part of sentence). The origins remain a point of debate that should be well-cited with appropriate recent literature. A few examples (there are many more) to consider citing are:

10.1038/s41588-021-00806-1;

10.1038/s41588-021-00818-x;

10.1016/j.devcel.2021.09.014;

10.1016/j.ccell.2020.08.014

These two reviews are also used for lines 82-87 for reasons that are unclear. It’s less of an issue in this second case, however the long sentence is difficult to understand – particularly what the first half (interplay between factors) has to do with the second (improper gene expression). A rewriting and reciting after a careful review of recent literature would be helpful.

5. Lines 340-342 (as one example): To draw such a direct conclusion as worded (“this event prevents them”), more than a schematic should be referenced (fig 7D). What are the primary data to support this point? Throughout this manuscript, care should be taken to either show direct effects experimentally or reword conclusions to be less certain or absolute. MYCN overexpression has broad effects, and correlation is not causation.

Additional/Minor Points:

1. Figure 4 J-K, is there a slight change in Phox2b levels? Hard to say without quantitation. Also, please add embryo ages.

2. “These results shed light on a potential mechanism behind MYCN overexpression and its effect on correct SAP differentiation by modulation of BMP signaling.” Please discuss – what exactly is the postulated mechanism?

3. Phox2b staining was accompanied by phox2bb HCR only. Is phox2ba not expressed?

4. Please consider using a grammar checking tool to improve readability/clarity for a broad non-expert audience.

6. PLOS authors have the option to publish the peer review history of their article (what does this mean?). If published, this will include your full peer review and any attached files.

Reviewer #1: No

Reviewer #2: No

---

## [Author Response · Author response to Decision Letter 0]

9 Aug 2024

Response to Reviewers and Editor

Manuscript title: Expansion of a neural crest gene signature following ectopic MYCN expression in sympathoadrenal lineage cells in vivo

ID: PONE-D-24-13321

We are grateful for the insightful comments and suggestions from the reviewers and have made edits to this manuscript to incorporate the recommendations. Please find below our detailed revisions and responses to the reviewer’s comments. In the revised manuscript, changes to address the comments below are in red text. 

Reviewer #1: Interesting paper that analyzes effects of MYCN expression on SA lineage development in zebrafish in vivo. Authors note that MYCN blocks differentiation, demonstrate in a candidate gene approach that BMP signaling is downregulated in MYCN cells, and then phenocopy aspects of differentiation block using a BMP inhibitor.

1. Authors describe markers in neural crest development in the introduction, but fail to mention Crestin, which is the main marker they study early in the paper.

We added a sentence in the introduction describing crestin as a marker for identifying NCCs in zebrafish: “In zebrafish, in addition to sox10, the crestin marker is expressed in premigratory and migratory NCCs, and its expression is gradually downregulated in differentiating cells (25).” 

2. In Fig 5, are the crestin elavl3 double positive cells and the elavl3 control cells proliferating?

Our previous results in Fig. 1 indicate that there are no significant changes in the number of SAPs found in the SCG of EGFP+ and EGFP-MYCN+ fish, suggesting this population is not robustly proliferative. To address the reviewer’s question and investigate if there were changes in proliferation between EGFP+ and MYCN+ cells, we performed WICHCRs against EGFP, crestin, elavl3, and phosphorylated Histone H3 at 4 dpf. We could only find one MYCN overexpressing fish (1/10) containing a proliferating crestin+/elavl3+ cell, indicating finding this aberrant (crestin+/elavl3+) population that was proliferating is rare. 

These findings are reflected in the text in the results section (lines 241-247) and supplementary figure (Supplementary Fig.5).

3. Can a structurally distinct inhibitor be used in addition to K02288 to assure the effects of this agent were on target? A rescue or genetic approach could also be used to achieve this.

To address this, we performed additional chemical inhibition assays with Dorsomorphin. After WICHCRs, we found that this inhibitor could also mimic the results found with K02288. Figure 6 has been updated to reflect Dorsomorphin treatment (fig. 6D-G). The results text has been updated accordingly to reflect Dorsomorphin treatments (lines 270-284).

4. Data on effects of BMPi are fairly preliminary, only 2 markers analyzed.

We performed additional WICHCRs against crestin and elavl3 and found that BMP inhibition led to a subtle increase in the expression of crestin in EGFP+ larvae treated with K02288 or Dorsomorphin, shown in supplementary figure (Supplementary Fig. 7). While this effect is not as strong as the one seen in sox10 expression (Fig. 6D-G), together, they highlight that BMP alterations are sufficient to result in aberrant NCC gene expression.

These new results are described in text in the results section (lines 285-289) and in supplementary figure (Supplementary Fig. 7).

Reviewer #2: In this manuscript, Padilla et al. propose that MYCN overexpression early in SAP development leads to abnormal gene expression of NCC, SAP, and neuronal markers. The authors observe this transient change through live imaging, suggesting a critical period of time during which MYCN affects NCC marker activation, and they evaluate crosstalk between MYCN overexpression and BMP signaling by measuring levels of phosphorylated-Smad and the BMP target id2a. Finally, they use publicly available scRNAseq datasets of NCC and NCC-derived cells to strengthen their findings.

Major Points:

1. One overarching concern is that the focus on spatiotemporal changes is not well-defined (timepoints or range of time) for much of the presented data and conclusions. Thus, the overall scientific advance, particularly compared to previously published work, is difficult to understand and interpret in context. 

Examples include:

a) Line 198: When does SAP differentiation begin and how is that defined? Please explain what is meant by (hpf) “very early SAP development”.

This sentence was reworded (lines 202-204) to make time points clearer: “These data indicate that sox10 reporter expression in EGFP-MYCN+ larvae is expanded transiently from 3 to 5 dpf.”

 Additionally, at the beginning of the results section, we added information on SAP development as it relates to when we first defined the SCG as our region of interest (lines 121-126); this way, the reader has a better understating of the timeline that comprises early SAP development and its markers and why we focused on these time points. 

b) Lines 98-101: “…it is known that MYCN expression is important for early NCC development… the role earlier ectopic MYCN overexpression has on NCC and SAP development has not been fully characterized…” While not necessarily contrary statements, this is confusing as written. Again, timepoints and a clearer statement about what the authors are referring to might help.

We greatly revised that sentence for clarity. We thank the reviewer for the suggestion.

c) Lines 238-240: “…MYCN overexpression is sufficient to severely alter the NCC to SAP cell differentiation gene expression program, across space and time, during early SAP development...” ‘Severely altering’ and ‘across space and time’ overstate conclusions and also appear to be at odds with the authors’ own ideas, given preservation of parts of the gene expression program and the emphasis throughout on specific time ranges (not across time), respectively.

We reworded the phrase to remove unnecessary text and make a more concise conclusion (top of page 11), included a range of time to put it under developmental context better: “MYCN overexpression is sufficient to alter SAP differentiation by promoting the expansion of NCC gene expression programs from 3 to 5 dpf, early timepoints during early SAP development.”

d) Figure 3 uses 70 hpf embryos. If SCG development starts around 48 hpf, please clarify why 70 hpf was the focus here.

The reviewer raises an important question. While dbh transcript is first detected at the onset of 2 days post fertilization (dpf), the earliest we could reliably identify and sort for embryos expressing dbh-driven EGFP/EGFP-MYCN within their SCG was at the transition between 2 and 3 dpf. Therefore, we focused our analyses on larvae beginning at 3 dpf. We have added this information to the beginning of the results section.

e) An in depth explanation of the previously known timing and regulation of MYCN's impact on SAP differentiation would be generally helpful.

We updated the introduction to reflect MYCN’s impact on development, in particular for nervous system development. Introduction page 5.

f) In Figure 4 (A-H), please justify/contextualize use of the 3 dpf timepoint.

The markers used in figure 4 are expressed during sympathoadrenal fate acquisition and are expected to be expressed by sympathetic fated cells at 3 dpf. We added a brief explanation as to why these time points were selected (line 209) to make it clearer to the reader.

g) Line 279: “…critical window…” Please define what the authors believe that to be.

Added the time point information to define the window we are describing in the discussion: “…a critical window, namely from 3 to 5 dpf, during which MYCN alters NCC marker activation.” 

h) Lines 296-299: “…our work expands spatiotemporal information regarding the transcriptional identity of these cells by unveiling the transient expansion…” Same request.

We corrected it similar to the comment above, where we added the exact time points we are referring. Updated discussion text is found on bottom of page 13.

2. The abstract states “dampened BMP signaling activity…suggesting BMP is functionally important for the NCC to SAP differentiation transition.” Disagree. Please modify this overstatement.

The abstract was modified to be less absolute. We removed the “and suggesting BMP is functionally important for the NCC to SAP differentiation transition.” to prevent any overstatements. (lines 32).

3. Lines 221-223: It appears that “does affect” should read “does NOT affect”.

This has been corrected, the sentence now reads “does not affect” (line 228). 

4. Lines 289-291: “Defective NCC specification towards SAP differentiation has been suggested as the origin mechanism for NB, where it is hypothesized that defects in cell fate acquisition promote a cell population that can form neoplastic lesions and give rise to NB (24,31).” There are several overlapping concerns here: Two reviews 5-6 years ago are used to justify this statement. Neither review appears to make this point, and there may be some field-specific confusion about terminology such has ‘origins’ and ‘derived from’. While NB is accepted as a ‘NC-derived cancer’, that is viewed by most of the field to mean origination from NC-derived sympathoadrenal progenitors/differentiated cells, not direct NC derivation as the authors appear to state (first part of sentence). The origins remain a point of debate that should be well-cited with appropriate recent literature. A few examples (there are many more) to consider citing are:

10.1038/s41588-021-00806-1;

10.1038/s41588-021-00818-x;

10.1016/j.devcel.2021.09.014;

10.1016/j.ccell.2020.08.014

These two reviews are also used for lines 82-87 for reasons that are unclear. It’s less of an issue in this second case, however the long sentence is difficult to understand – particularly what the first half (interplay between factors) has to do with the second (improper gene expression). A rewriting and reciting after a careful review of recent literature would be helpful.

We appreciate the reviewer bringing this to our attention, the discussion text has been updated (2nd paragraph of discussion). 

The sentence in the introduction has been removed.

5. Lines 340-342 (as one example): To draw such a direct conclusion as worded (“this event prevents them”), more than a schematic should be referenced (fig 7D). What are the primary data to support this point? Throughout this manuscript, care should be taken to either show direct effects experimentally or reword conclusions to be less certain or absolute. MYCN overexpression has broad effects, and correlation is not causation.

We thank the reviewer for raising this question, we have reworded the conclusion to make it less absolute and reflect that we are postulating a potential model of action (page 15). This is a hypothetical explanation to the result we are seeing and an exact interaction between MYCN and BMP signaling should be studied further in future studies.

Additional/Minor Points:

1. Figure 4 J-K, is there a slight change in Phox2b levels? Hard to say without quantitation. Also, please add embryo ages.

Embryo ages have been added to Figure 4J-K, and total Phox2b expression in the SCG quantified. We did not detect a significant change in Phox2b levels between conditions.

2. “These results shed light on a potential mechanism behind MYCN overexpression and its effect on correct SAP differentiation by modulation of BMP signaling.” Please discuss – what exactly is the postulated mechanism?

A brief description of the proposed model was added in lines at the end of the results section (lines 291-296).

3. Phox2b staining was accompanied by phox2bb HCR only. Is phox2ba not expressed?

We did not analyze the expression of phox2ba, so we do not know if phox2ba is expressed. However, based on the other sympathetic/neuronal markers used, we can be confident that we are detecting sympathetic fated cells with these markers used.

3. Please consider using a grammar checking tool to improve readability/clarity for a broad non-expert audience.

Grammarly was used to help identify grammatical errors and correct them throughout the text.

---

## [Decision Letter · Decision Letter 1]

27 Aug 2024

Expansion of a neural crest gene signature following ectopic MYCN expression in sympathoadrenal lineage cells in vivo

PONE-D-24-13321R1

Dear Dr Uribe,

We’re pleased to inform you that your manuscript has been judged scientifically suitable for publication and will be formally accepted for publication once it meets all outstanding technical requirements.

Kind regards,

Tao Liu, PhD

Academic Editor

PLOS ONE

Additional Editor Comments (optional):

Reviewers' comments:

Reviewer's Responses to Questions

**Comments to the Author**

1. If the authors have adequately addressed your comments raised in a previous round of review and you feel that this manuscript is now acceptable for publication, you may indicate that here to bypass the “Comments to the Author” section, enter your conflict of interest statement in the “Confidential to Editor” section, and submit your "Accept" recommendation.

Reviewer #1: All comments have been addressed

Reviewer #2: All comments have been addressed

2. Is the manuscript technically sound, and do the data support the conclusions?

Reviewer #1: Yes

Reviewer #2: (No Response)

3. Has the statistical analysis been performed appropriately and rigorously? 

Reviewer #1: Yes

Reviewer #2: (No Response)

4. Have the authors made all data underlying the findings in their manuscript fully available?

Reviewer #1: Yes

Reviewer #2: (No Response)

5. Is the manuscript presented in an intelligible fashion and written in standard English?

Reviewer #1: Yes

Reviewer #2: (No Response)

6. Review Comments to the Author

Reviewer #1: (No Response)

Reviewer #2: (No Response)

7. PLOS authors have the option to publish the peer review history of their article (what does this mean?). If published, this will include your full peer review and any attached files.

Reviewer #1: **Yes: **William A. Weiss

Reviewer #2: No

---

## [Editor Report · Acceptance letter]

8 Sep 2024

PONE-D-24-13321R1 

PLOS ONE

Dear Dr. Uribe, 

I'm pleased to inform you that your manuscript has been deemed suitable for publication in PLOS ONE. Congratulations! Your manuscript is now being handed over to our production team.

Kind regards, 

on behalf of

Dr. Tao Liu 

Academic Editor

PLOS ONE